# Younger Age in Adolescent Pregnancies Is Associated with Higher Risk of Adverse Outcomes

**DOI:** 10.3390/ijerph18168514

**Published:** 2021-08-12

**Authors:** Maria de la Calle, Jose L. Bartha, Cristina M. Lopez, Miriam Turiel, Nuria Martinez, Silvia M. Arribas, David Ramiro-Cortijo

**Affiliations:** 1Obstetrics and Gynecology Service, Hospital Universitario La Paz, Paseo de la Castellana 261, 28046 Madrid, Spain; maria.delacalle@uam.es (M.d.l.C.); joseluisbartha@me.com (J.L.B.); mturiel93@gmail.com (M.T.); nuriki@gmail.com (N.M.); 2Pediatrics Service, Hospital Universitario La Paz, Paseo de la Castellana 261, 28046 Madrid, Spain; cristina.lopez.1995@hotmail.com; 3Department of Physiology, Faculty of Medicine, Universidad Autónoma de Madrid, C/Arzobispo Morcillo 2, 28049 Madrid, Spain; silvia.arribas@uam.es; 4Department of Medicine, Beth Israel Deaconess Medical Center-Harvard Medical School, 330 Brookline Avenue, Boston, MA 02215, USA

**Keywords:** adolescent pregnancy, maternal complications, fetal complications, labor complications, obstetrical control

## Abstract

Adolescent pregnancy remains a health issue worldwide also in developed countries, since it has been associated with adverse maternal and neonatal outcomes. Some data suggest that very young adolescents have higher risk, likely due to immaturity. Therefore, we aimed to assess the influence of maternal age on complications during gestation and labor in pregnant women between 13 and 19 years of age. In particular, we evaluated the possible association between maternal age and obstetric, fetal and labor complications. This is a retrospective, observational and exploratory study conducted at Hospital Universitario La Paz (HULP, Madrid, Spain). The clinical history of 279 women who delivered between 2013 and 2018 was analyzed. Maternal age and the presence of maternal, fetal and labor complications, as well as risk of postpartum depression and breastfeeding intention, were recorded. General regression models were used to analyze the contribution of maternal age on each complication. The percentage of adolescent pregnancies at HULP between 2013 and 2018 was 0.9%. The risk of all the maternal complications analyzed decreased significantly with every year of age of the mother (hyperemesis, lower back pain, anemia, gestational diabetes mellitus, and threat of premature labor and premature rupture of membranes). Every year of maternal age decreased 0.8-fold [0.8; 0.9] the prevalence of fetal complications and also reduced the risk of C-section, postpartum hemorrhage and obstetrical hysterectomy. Furthermore, higher maternal age increased 1.1-fold [1.0; 1.2] the breastfeeding intention. In conclusion, young adolescents are at higher risk of complications during pregnancy and labor.

## 1. Introduction

Adolescent pregnancy is a global health problem affecting different countries at varying degrees. Although a global decline in births has been found since the implementation of contraceptive measures and legalization of abortion in many countries, there are still many women who become pregnant between 13 and 19 years old. In developing countries, approximately 19% of women below 19 years old become pregnant [1]. The highest rates are found in Africa, Central America, and South America [1]. Several sociocultural factors contribute to adolescent pregnancies, including low economic and educational level, growing up in dysfunctional family, family history of adolescent pregnancy, premature onset of sexual activity, early menarche and lack of information on the use of contraceptive measures [2,3,4]. In addition, higher rates are also found in some societies where adolescent pregnancies are an intentionally marital event and are not considered a social problem [5,6].

From the medical point of view, adolescent pregnancies have a higher risk of complications and there is evidence that maternal mortality in the 13–19 age group is two times more compared to the 20–34 age group [7]. Regarding maternal complications, higher rates of anemia, pregnancy-induced hypertension and preeclampsia have been reported [8]. There is also evidence of higher rates of fetal complications, such as fetal growth restriction (FGR), prematurity, miscarriage, and fetal death [9,10]. In addition, during labor, adolescent pregnancies have been associated with instrumented deliveries [11]. Adolescent pregnancies also have long-term problems, such as higher rates of maternal postpartum depression, which influence maternal-neonatal bound and reduce adherence of breastfeeding [12] and emotional syndrome in the offspring [13,14]. All these complications in adolescents could be worsened by maternal malnutrition, toxic habits and inadequate prenatal care [6,7,13,14]. Therefore, the World Health Organization (WHO) warns of the importance of caring for pregnant adolescents, especially in underdeveloped and developing countries, which have poorer health systems [13,15].

In high-income countries, the rates of adolescent pregnancies are lower, due to socioeconomic and cultural factors. However, they are still high in some settings. In Spain, the adolescent fertility rate in women between 15–19 years old is 0.73% [16], despite the sociocultural level, implementation of sex education programs, and availability of contraceptive information. Some data evidenced that the rate of unplanned adolescent pregnancies can reach 29.8% in Spain [17]. Even in settings with a consolidated health system, adolescent pregnancies are not devoid of complications. Therefore, it is necessary to expand the knowledge on the risk of maternal-fetal complications in adolescent pregnancies and their determinants.

Data from developing countries indicate that fetal and obstetric complications are higher in adolescents with ages near menarche [18,19,20]. Poor socioeconomic conditions or health systems may have exerted an influence and may be confounding factors. Therefore, it is important to evaluate the possible influence of maternal age on adolescent pregnancies complications, in other settings with higher socioeconomic level and robust public health systems. Therefore, our aim was to analyze the possible influence of maternal age on different types of complications during gestation and labor in pregnant women between 13 and 19 years of age in Spain, a high-income country and with a well-established public health system, accessible for every citizen.

## 2. Materials and Methods

### 2.1. Sample of the Study

This is a single-center retrospective, non-interventional and observational study from the Obstetrics and Gynecology department in the Hospital Universitario La Paz (HULP, Madrid, Spain). The maternity of this hospital is one of the largest in Spain, attending annually approximately 5500 deliveries. The cohort of the study were women who attended at the hospital, between January 2013 to December 2018. The inclusion criteria was pregnant women with age between 13 to 19 years old, over 22 weeks of gestational age, who had obstetric follow-up and labor at HULP. The exclusion criteria were: Incomplete medical data. Since we focused on maternal age as a risk factor, we excluded women with diagnoses of sexually transmitted infections (n = 5) or toxic habits during pregnancy (n = 13, including smoking, alcohol and/or illegal drug intake) to eliminate possible confounding factors influencing the risk of complications. Finally, a total of 279 women matched with the inclusion criteria and were included in this study.

The rates of adolescent pregnancies in the years included in the study were: 2013 = 0.9% (51/5710); 2014 = 0.9% (51/5607); 2014 = 0.8% (43/5648); 2016 = 0.9% (50/5671); 2017 = 1.1% (58/5469); and 2018 = 0.9% (46/5178). 

According to the age of conception, women were categorized in two groups: Below 17 years old (n = 131) and equal to 17 years or older (n = 148). We established these two groups considering the high risk of obstetric complications in women with pregnancies in the first 5 years after menarche [18], assuming an average menarche age of 11 years [21].

This study was performed in accordance with the Declaration of Helsinki regarding studies in human subjects and it was approved by HULP Ethical Committees (PI-4287). 

### 2.2. Data Collection

Data were retrospectively obtained from medical records. All the women attended at HULP go through the same procedures, with regular visits. The data were collected during follow-up visits and completed during labor by the obstetricians. Some women (10.8%) did not have prenatal health care and only went to the hospital for the labor. Therefore, in these women only delivery data could be recorded. The diagnosis of the complications was performed by the medical staff according to Hospital Guidelines

**Maternal variables during pregnancy**. Age (years), ethnicity (categorized in Caucasian, Latin, and Others), parity (delivery before), type of gestation (single/twin), obstetric control (means one visit monthly with the obstetrician or the midwife), comorbidities (any disease during the previous 5 years including obesity, hypertension, diabetes mellitus, cancer, immune diseases, hematologic diseases, rheumatologic diseases, and endocrinology diseases). The reproduction variable was also included as a type of conception: Spontaneous or through assisted reproduction techniques (ART). Gestational age at delivery (weeks) was also recorded.

**Maternal-Obstetrical complications.** The following complications were included, if they occurred at any time during gestation. Hyperemesis, lower back pain (presence of pain localized to the lower back area or radiating into the buttock, thigh, and legs, causing or mimicking sciatica symptoms), gastroesophageal reflux (presence of heartburn), anemia (hemoglobin lower than 11 g/dL), gestational diabetes mellitus (a positive result in the 100 g oral glucose tolerance test), pregnancy induced hypertension (systolic blood pressure higher than 140 mmHg and/or diastolic blood pressure higher than 90 mmHg after 20 weeks of gestational age), preeclampsia (systolic blood pressure higher than 140 mmHg and/or diastolic blood pressure higher than 90 mmHg and proteinuria over 300 mg in 24-h urine), threat of preterm labor, (contractions and/or cervical change before 37 weeks), premature rupture of membranes (loss of integrity of the amniotic membranes before the onset of labor). To obtain maternal-obstetrical complications variable, women were categorized according to the presence (“yes”) of one or more of the above mentioned maternal-obstetrical complications. Otherwise, they were categorized as absence (“no”).

**Fetal complications.** The following complications were included, if they occurred at any time during gestation. Fetal malformation, assessed by echography (including urogenital tract, circulatory system, and nervous system), fetal growth restriction (FGR; fetal growth below the third percentile or below the tenth percentile with hemodynamic alterations), small for gestational age (SGA; fetus below the tenth percentiles without Eco-Doppler abnormalities), and fetal death. To obtain fetal complications variable, women were categorized according to the presence (“yes”) of one or more diagnostics of malformations, FGR, SGA or fetal death. Otherwise, they were categorized as absence (“no”).

**Labor complications**. Vaginal tear (categorized as grades 3 and 4, which is considered severe), postpartum hemorrhage, obstetric hysterectomy, and maternal death. Premature delivery (birth before 37 weeks of gestation) as prematurity, was also recorded. 

The route of delivery (vaginal/C-section) was also collected including the reason for C-section (pelvic-cephalic disproportion, podalic presentation, loss of fetal well-being defining by umbilical artery Doppler velocimetry [22], stationary delivery, previous C-section). To obtain labor complications variable, women were categorized according to the presence (“yes”) with one or more diagnostics of prematurity, C-section, vaginal tear, postpartum hemorrhage, obstetric hysterectomy, and maternal death. Otherwise, they were categorized as absence (“no”).

**Other variables studied**. In addition to the above-mentioned variables related to maternal-fetal complications, we also collected data regarding the breastfeeding intention (directly question to the mother); and screening on possible development of postpartum depression, through the Edinburgh Postpartum Depression Scale (EPDS), defined as cut-point ≥11 score [23]. 

### 2.3. Statistical Analyses

Normality was verified by the Kolmogorov-Smirnov test. Quantitative variables were summarized as median and interquartile range [Q1; Q3] and qualitative variables were expressed as absolute (n) and relative frequencies (%). Differences between groups were assessed via Mann-Whitney’s U-test for quantitative variables, and associations were tested by the chi-squared test. Spearman’s rho (*ρ*) coefficient was used to test correlations in quantitative variables. To test the contribution of increasing 1 year in maternal age on each complication, general regression models were built, adjusted by ethnicity, obstetric control and parity. The adjusted odds ratio (aOR) and 95% confidence interval [CI] were calculated and associated *p*-Values were extracted. Significance probability was established at *p*-Value < 0.05. Statistical analysis was performed with R software (version 3.6.0, 2018, R Core Team, Vienna; Austria) within R Studio interface using *rio, tidyverse, dplyr, questionr, ggplot2*, and *ggpubr* packages.

## 3. Results

The median of maternal age was 17.0 [16.0; 17.0] years old. The most representative ethnic group was Caucasian (61.3%; 171/279), being Romanian the largest cluster (93.6%; 160/171). The second ethnic group was formed by women of Latin-American origin (35.5%; 99/279). There was no previous comorbidities in any of the women studied. 

During pregnancy, 89.2% (240/279) of the women had obstetric control and labor assistance, and 8.6% (24/279) of the adolescents had a previous labor. All pregnancies were singleton and with a spontaneous conception. 

The median of gestational age was 39.4 [38.1; 40.2] weeks. There were no stillbirths in the cohort. Gestational age at delivery was lower in the younger group, but did not reach statistical significance and gestational age at delivery was not significantly correlated with maternal age (rho = 0.05; *p*-Value = 0.40).

Table 1 shows the data in the groups of study (below and over 17 years old) regarding maternal age, ethnicity, obstetric control, parity and gestational age at delivery. No significant differences between groups were observed in any of the variables studied.

Overall, maternal-obstetrical complications represented 25.4% (71/279), being premature rupture of membranes the most common. There was no evidence of preeclampsia in the cohort. 

Table 2 shows the prevalence of maternal-obstetric complications and differences between the groups of study. The maternal-obstetric adverse events per year of maternal age was not significantly different between groups (aOR = 0.9 [0.9; 1.0]; *p*-Value = 0.06). However, per each year of increase in maternal age, the risk of hyperemesis, lower back pain, anemia, gestational diabetes mellitus, threat of premature labor, and premature rupture of membranes significantly decreased (Table 2).

Fetal complications represented 5.4% (15/279) of the total cohort, being fetal malformations the most common, including those of the urogenital tract (such as renal pelvic dilatation and hypospadias), circulatory system (such as aberrant right subclavian artery and ventricular septal defect), and nervous system (choroid plexus cysts). There was no fetal death in the cohort. Table 3 shows the prevalence between groups, according to maternal age. Every year of maternal age decreased 0.8-fold [0.8; 0.9] the prevalence of fetal complications (*p*-Value = 0.001). Furthermore, per each year of increase in maternal age, the risk of malformations, FGR, and SGA also significantly decreased (Table 3). Our data indicate that higher maternal age decreased the risk of fetal complications in adolescent pregnancies.

Labor complications represented 55.2% (154/279) of the cohort, being vaginal tear the most prevalent. C-section represented 13.6% being the main reasons: Pelvic-cephalic disproportion (28.9%), C-section previously (26.3%), loss fetal well-being (18.4%), fetal podalic presentation (18.4%), and stationary delivery (7.9%). There was no maternal death during labor in the cohort.

Maternal and fetal complications tend to decrease in older adolescents. However, the opposite was found regarding labor complications, since we found that for each year in maternal age, the risk increased 1.1-fold [1.0; 1.1] (*p*-Value = 0.022). On the other hand, per each year of age the risk of C-section, postpartum hemorrhage, and obstetric hysterectomy significantly decreased. Table 4 shows the prevalence and differences in labor complications between groups. 

During the post-labor period, the overall intention of breastfeeding was 92.8% (259/279). The older maternal age increased 1.1-fold [1.0; 1.2] breastfeeding intention (*p*-value = 0.002). We did not find evidence of postpartum depression. It is important to note that the EPDS was used as a screening tool.

## 4. Discussion

The main finding of the present study is the evidence of the important influence of maternal age in the development of medical adverse outcomes in adolescent pregnancies. We demonstrated that in the per year increase of maternal age, there is a reduction in the risk of maternal and fetal complications, but an increase in the risk of labor complications. In addition, the intention of breastfeeding was also lower in younger adolescents. In our study, the majority of adolescent pregnancies were of non-Spanish origin. The present data stress the importance of providing specific follow-up and counselling in the adolescent population from a very early age, and to implement specifically sexual education politics, particularly designed for the immigrant population. 

Adolescent pregnancies are still elevated in high-income countries. The USA is one of the countries with highest rates (1.9%), being Latin and Black women the ethnic groups with highest prevalence, likely related to cultural and socioeconomic factors [14,24]. In Europe, the pregnancy rate in adolescent women was 1.0%. The Netherlands, Switzerland, Japan, and South Korea report lower rates [14], being Switzerland the lowest (0.3%) of Europe and Korea (0.03%) in the world [1]. In Spain, the rates established in 2017 were 0.6% being in the bottom of the list compared to other countries in the European Union. According to the data in the present study, the adolescent pregnancy rate has remained unchanged for five years, being 0.9% between 2013 and 2018.

In our sample, the most represented ethnic group was Caucasian women (over 61%; most of them from Romania), followed by women from Latino-American origin (35%). Thus, adolescent pregnancies in Spain are mostly from women of non-Spanish origin. This may reflect their origin background. It has been described that high adolescent pregnancy rates are in Nicaragua, Venezuela, Panama, Ecuador, and Romania [15]. In these countries, the adolescent fertility rate was around 3.3% and remained constant from 2000 [25]. According to the Spanish Institute of Statistics [26] regarding the prevalence of pregnancies in Spain in women below 15 years old, the rate was 0.1%, while the rate in non-Spanish women was 0.6%. Pregnancy rates are also higher in women of 18 years old (0.8% in Spanish women and 3.0% non-Spanish women). As age increases, the adolescent pregnancies rates increase in both groups, but in non-Spanish women, the ratio is more than three times up. These rates have remained constant in recent decades [27] despite access to sexual health information and contraceptive methods in Spain. Therefore, our data suggest that the migratory flow plays an essential role in the data on adolescent pregnancies, as previously described [15], and that sociocultural factors from their origin background have prevalence over those of Spanish society. It is also possible that these women have less access to information and education. According to the literature, close to 75% of the adolescents with history of pregnancy had an educational gap. In addition, these educational gaps were associated with multiparous women, low socio-economical level. On the other hand, school assistance seems to be a protective factor [28]. For this reason, it would be necessary to implement effective and efficient educational public politics, to improve sexual education in the school system preventing adolescent pregnancy. In respect, considering socio-cultural factors to reach the entire population. Another factor is the access to information about contraceptive methods. Oral contraceptives are available in Spain from the 60’s and its use is now generalized. Despite some debate about its adverse effects, its use is well accepted by the community [29]. In addition, the Spain health system has family planning units where the population can be well informed about its use. However, it is possible that the population from other cultural settings has less access to information or more prejudices about contraceptive methods. Therefore, sex education in adolescents and the access to these information for the immigrant population needs to be revised. Government policies should ensure culturally sensitive medical care as particularly important in adolescent pregnancies. In HULP, around the third trimester of adolescent pregnancies are followed-up for contraceptive counseling. Long-acting subdermal implants are the preferred choice which are offered free to these women in our national health system.

Regarding to medical problems in adolescent pregnancies, it has been demonstrated that complications for these young women and their children could happen more frequently, both during pregnancy and postpartum, being at greater risk those who become pregnant in the first 5 years after menarche [18]. This study found a relationship between maternal complications and the younger age group, particularly in hyperemesis, lower back pain, anemia, gestational diabetes mellitus, and threat of preterm labor and premature rupture of membranes. The severe complications such as anemia, gestational diabetes mellitus, and threat of preterm labor rates were lower compared to the general pregnant population. It is known that these complications are more related to the old childbearing age [30,31], considering advanced maternal age (more than 35 years old). By contrast, the prevalence in our cohort of premature rupture of membranes (9.3%) was higher than of the general population (2–8%) [32,33,34,35]. It has been described that the premature rupture of membranes is more frequent in younger adolescents with low socioeconomic status [36,37]. It would be necessary to assess whether nationality could also be related to socioeconomic status. A low socioeconomic status has been associated with poor health conditions [38], potentially mediated by nutrition misbalance and toxic habits [39]. All these factors could bias adverse medical outcomes such as premature rupture of membranes in adolescent pregnancies. 

Our study found that the increasing maternal age was a protective factor to reduce the risk of developing fetal complications, being the most frequent in our cohort urogenital tract, cardiac and nervous systems fetal malformations. It has been shown that some malformations are more common in adolescents, such as gastrointestinal and central nervous system anomalies [14]. Although rates of FGR and SGA have been described as higher in the general pregnancies population than our sample [32,33,34,35], it is possible that the rate of fetal complications may be higher than observed since close to 11% of the recorded pregnancies were not under obstetrical control. Poor obstetrical follow-up is one of the most relevant risk factors for maternal-fetal complications, particularly important in adolescents [36,40]. Therefore, higher morbi-mortality was observed in adolescent pregnant women and their children, since the associated medical problems are detected late [41]. Thus, all our associations were adjusted by obstetrical control. Rejection of pregnancy, fear of the parents’ reaction and lack of knowledge about pregnancy have been proposed as variables that impede obstetric control in adolescent pregnant women [24]. It known that if prenatal care is adequate, the pregnancy outcome in adolescents over 15 years old are comparable to pregnancies in adult women [42], and reduce obstetrical complications [2]. Therefore, it is important to promote obstetrical control from the earliest stages of pregnancy, since it has been shown that the social context, associated with inadequate prenatal follow-up, raise the risk of maternal-fetal complications even more than the biological immaturity of the woman [2,4,37,43].

Previous data have evidenced that adolescents have higher risk of C-section and need of forceps compared to non-adolescents [13]. In our population, the majority of pregnancies had vaginal delivery, and 13.6% had C-sections. The position of the fetus and the incomplete development of the maternal pelvis could determine an inability of the birth canal to allow the passage of the fetus; these dystocias cause an increase in operative deliveries (forceps and C-sections). According to our data, the main reason for C-section was pelvic-cephalic disproportion. This is more common in women below 15 years old due to the narrower pelvis of adolescent women, as indicated by other authors [44]. According to our data, older maternal age could protect against postpartum hemorrhage and hysterectomy. Surprisingly and overall, complications during labor increased with maternal age. The rate of vaginal tear (32.3%), C-section (13.6%) and prematurity in our sample (11.1%, which can be related) could increase the rate of complications during labor. Although it was not significant, we detected a trend association between the high prematurity rate and increase of maternal age. The rate of prematurity in our sample was higher compared to the general population [32,33,34,35]. Other authors also found high prematurity rates in adolescent pregnancies [3,6].

Our data demonstrate that the risk of maternal-fetal complications in adolescents are higher in those of younger age. An early diagnosis is essential, focusing on adequate control to reduce complications. In Spain, adolescent pregnant women are considered a risk group and are attended into their reference hospital from Primary Health Centers. However, some women may not attend the health centers during pregnancy, due to fear or ignorance and will arrive to the hospital only for labor. In fact, in the present study we observed this and considering that younger women have higher risk of most complications during pregnancy, it would be important to have a closer obstetrical follow up in this age group.

In addition to obstetric complications, we also aimed to analyze some long term parameters, including the risk of postpartum depression and low rates of breastfeeding, since adolescent mothers have higher risk and these two factors are closely related. We evidenced lower breastfeeding adherence in relation to the age of the mother, being lower in younger adolescents. Reduced breastfeeding intention has been associated with maternal postpartum depression [12]. However, in our sample, we did not observe postpartum depression, assessed through EPDS. The risk of postpartum depression in adolescents, has been associated with social difficulties and of toxic substances abuse [2]. In this study, we excluded women using toxic substances. In addition, the risk of postpartum depression may extend up to six months after labor [12], and we did not have the opportunity to follow the mothers since this was a retrospectives study. Therefore, we cannot exclude that lower breastfeeding adherence and postpartum depression in our study group may be related. Future studies are required to analyze this aspect and to evaluate the need to implement counselling programs for adolescent women through multidisciplinary teams, as previously suggested [45].

We also showed that the majority of adolescent pregnant women were of non-Spanish origin, suggesting an important influence of immigration. Our population was obtained in Madrid, a multicultural city and may not reflect the situation of other areas in Spain with less immigration. This higher level of adolescent pregnancies in immigrants may be related to a lower socio-economic and educational status, since a poor background usually brings school abandonment, high levels of stress, and family conflicts [7], factors associated with an early pregnancy. From a social point of view, it would be desirable to implement sex education for adolescents, not only in a school setting, but also at the primary care level, including the use of contraceptive measures and the promotion of adequate socio-family support. 

As a general conclusion, our data evidencing the importance of maternal age in the risk of complications and breastfeeding adherence in adolescent pregnancies, support the need to implement health strategies to reduce the risks, such as improving their prenatal assistance and postnatal follow-up through multidisciplinary teams, and other specific policies recognizing their higher risk.

### Study Limitations and Research Perspective

Firstly, it needs to be considered that the percentage of adolescent pregnancies was low. However, the data could reflect the social changes in a European developed country with a multicultural perspective. The contribution of different levels of educational and socioeconomic spheres could modulate the results. Related to the above, it would be desirable to explore the economic level, family situation, level of education, among other variables, to study whether there are other risk factors independent of the mother’s age.

Secondly, smoking, alcohol, and illegal drug use are more frequent in adolescents, and it has serious repercussions on pregnancy, being susceptible to FGR and premature rupture of membranes [40,44,46]. In our analysis, obtaining associations between maternal age and medical complications without other confounding factors were excluded. However, exploring the toxic habits in adolescent pregnancies would help to better understand the situation and evaluate the contribution to pregnancy complications. On the other hand, it is important to note that about 11% did not have an adequate obstetrical control. If there is no information about one complication, it does not necessarily mean that it did not occur, since it may not have been recorded. This is an important source of bias to be considered in our work.

On the other hand, it has been described that the younger the adolescent pregnant women, the greater the likelihood of a subsequent adolescent pregnancy [7], mainly due to the lack of use of contraceptive methods [11]. More studies would be required to verify this hypothesis and the influence by the socioeconomic and educational context. In addition, our results indicate that younger maternal age is linked to higher risk of obstetrical complications, likely related to immaturity. In order to confirm this, it would have been desirable to know the age of menarche of the women, which was not possible to assess in this study.

## 5. Conclusions

Adolescent pregnancy continues to be a relevant health issue today in developed countries, being imperative to implement strategies to control and prevent it. In this study, we have demonstrated that the youngest adolescents have higher risk for complications for the mother and child. Our data also evidence that the majority of adolescent pregnancies occur in women of non-Spanish origin, which may have lower socioeconomic and educational level. Therefore, it is necessary to establish specific interventions for these groups, such as sex education programs and contraceptive methods, in family, schools as well as in primary health care centers. It is also necessary to obtain information about other the risk factors associated with the risk of complications in adolescent pregnancy, such as the use of drugs. Furthermore, it would be desirable to establish health policies to derive pregnant adolescents to the reference hospitals earlier, for a closer follow-up and reinforcement of reinforcement the psychological sphere.

## Figures and Tables

**Table 1 ijerph-18-08514-t001:** Maternal variables during pregnancy according to maternal age.

**Maternal Variables**	**Below 17 Years Old** **(n = 131)**	**Over 17 Years Old** **(n = 148)**	***p*** **-Value**
Maternal age (years)	16.0 [15.0; 16.0]	17.0 [17.0; 17.0]	<0.001 ^a^
Ethnicity			0.25 ^b^
Caucasian	66.4% (87)	56.8% (84)	
Latin	30.5% (40)	39.9% (59)	
Others	3.1% (4)	3.4% (5)	
Obstetric control	82.4% (108)	89.2% (132)	0.15 ^b^
Parity (multiparous)	6.1% (8)	10.8% (16)	0.24 ^b^
Gestational age at delivery (weeks)	39.3 [37.9; 40.0]	39.6 [38.3; 40.4]	0.06 ^a^

Data show median and [Q1; Q3] or percentage and sample size (n). *p*-Value was extracted by ^a^ Mann-Whitney’s U-test and ^b^ chi-squared test as appropriate.

**Table 2 ijerph-18-08514-t002:** Maternal-obstetrical complications during pregnancy according to maternal age.

Maternal and Obstetric Complications	Below 17 Years Old (n = 131)	Over 17 Years Old (n = 148)	*p*-Value ^a^	aOR [95% CI]	*p*-Value ^b^
**Maternal-Obstetrical complications**	26.0% (34)	25.0% (37)	0.96	0.9 [0.9; 1.0]	0.06
Hyperemesis	7.6% (10)	7.4% (11)	0.99	0.9 [0.8; 0.9]	<0.001
Lower back pain	0.0% (0)	2.0% (3)	0.29	0.8 [0.5; 0.9]	0.008
Gastroesophageal reflux	2.3% (3)	1.4% (2)	0.89	0.7 [0.3; 0.8]	0.08
Anemia	2.3% (3)	6.1% (9)	0.21	0.8 [0.7; 0.9]	<0.001
Gestational Diabetes Mellitus	2.3% (3)	0.7% (1)	0.53	0.8 [0.7; 0.9]	0.001
Pregnancy induced hypertension	0.8% (1)	0.7% (1)	0.99	0.4 [0.1; 1.6]	0.21
Preeclampsia	0.0% (0)	0.0% (0)	-	-	-
Threat of preterm labor	3.1% (4)	3.3% (5)	0.99	0.8 [0.7; 0.9]	<0.001
Premature rupture of membranes	11.5% (15)	7.4% (11)	0.34	0.9 [0.8; 0.9]	<0.001

Data show the percentage and sample size (n). ^a^
*p*-Value was extracted by the chi-squared test. Adjusted odds ratios (aOR), 95% confidence intervals [95% CI], and ^b^
*p*-Value associated was extracted from general regression models adjusted for ethnicity, obstetrical control and parity. The regression models were built considering age as a continuous variable.

**Table 3 ijerph-18-08514-t003:** Fetal complications during pregnancy according to maternal age.

Fetal Complications	Below 17 Years Old (n = 131)	Over 17 Years Old (n = 148)	*p*-Value ^a^	aOR [95% CI]	*p*-Value ^b^
**Fetal complications**	3.1% (4)	7.4% (11)	0.19	0.8 [0.8; 0.9]	0.001
Malformations	2.3% (3)	3.4% (5)	0.60	0.9 [0.8; 0.9]	0.003
Fetal growth restriction	0.8% (1)	3.4% (5)	0.14	0.7 [0.5; 0.9]	0.005
Small for gestational age	0.8% (1)	2.0% (3)	0.38	0.8 [0.5; 0.9]	0.012
Fetal death	0.0% (0)	0.0% (0)	-	-	-

Data show the percentage and sample size (n). ^a^
*p*-Value was extracted by the chi-squared test. Adjusted odds ratios (aOR), 95% confidence intervals [95% CI], and ^b^
*p*-Value associated was extracted from general regression models adjusted for ethnicity, obstetrical control and parity. The regression models were built considering age as a continuous variable.

**Table 4 ijerph-18-08514-t004:** Labor complications during pregnancy according to maternal age.

Complications during Labor	Below 17 Years Old (n = 131)	Over 17 Years Old (n = 148)	*p*-Value ^a^	aOR [95% CI]	*p*-Value ^b^
**Labor complications**	56.5% (74)	54.1% (80)	0.77	1.1 [1.0; 1.1]	0.022
Prematurity	12.2% (16)	10.1% (15)	0.58	1.0 [0.9; 1.0]	0.06
C-section	16.0% (21)	11.5% (17)	0.29	0.9 [0.9; 1.0]	0.003
Vaginal tear	30.8% (40)	34.0% (50)	0.57	1.0 [0.9; 1.0]	0.21
Hemorrhage	3.1% (4)	1.4% (2)	0.33	0.8 [0.7; 0.9]	<0.001
Obstetric hysterectomy	0.8% (1)	0.0% (0)	0.29	0.7 [0.5; 0.9]	0.018
Maternal death	0.0% (0)	0.0% (0)	-	-	-

Data show the percentage and sample size (n). ^a^
*p*-Value was extracted by the chi-squared test. Adjusted odds ratios (aOR), 95% confidence intervals [95% CI], and ^b^
*p*-Value associated was extracted from general regression models adjusted for ethnicity, obstetrical control and parity. The regression models were built considering age as a continuous variable.

## Data Availability

The data presented in this study are available on request from the corresponding author. The availability of the data is restricted to investigators based in academic institutions.

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
