# Peer review of "Younger Age in Adolescent Pregnancies Is Associated with Higher Risk of Adverse Outcomes"

_ijerph, 2021, doi:10.3390/ijerph18168514_

Round 1

Reviewer 1 Report

observations:

  1. line 104-obstetric control (regular obstetric visit and labour assistance ) -need to be more specific what does it mean regular obstetric visit and labour assistance and how it was quantified ??
  2. line 110-what does it mean lower back pain (detail) and how it was quantified the lower back pain and gastroesophageal reflux ??
  3. line 127-vaginal tear - to be more specific grade 1, grade 2 etc specifically grad 3 and 4 ???
  4. line 137- 138 and 210 -it must be point out that EPDS it is not an diagnostic tool but a screening tool so you must introduce the results in that context 
  5. line 131- losss of fetal well being -must be more precisely point out -variable decelerations, late decelerations, bradycardia  etc ????
  6. finally something must be point out in table 2 and table 3. There are some confusing data. line 178 and line 190 point out something and in  In table 2 the lower back pain is increased >17 years old  and not decreased , anemia is also increased in > 17 y old not decreased , and in table 3 the FGR is increased  in > 17 y old patients, SGA is also increased and not decreased in > 17 y old patient and overall malformations the same . So it must be detailed why are these results presented because also the p results at table 3 are confusing.

Author Response

Response: We would like to thank the reviewer for the time spent on our manuscript as well as for the comments and suggestions made. We hope that we have kindly answered all the queries.

Line 104-obstetric control (regular obstetric visit and labour assistance) -need to be more specific what does it mean regular obstetric visit and labour assistance and how it was quantified?

Response: Regular obstetric visit means once a month, with the obstetrician or the midwife (line 116). Labor assistance means that labor was assisted at the hospital.

Line 110-what does it mean lower back pain (detail) and how it was quantified the lower back pain and gastroesophageal reflux?

Response:  Lower back pain is a pain localized to the lower back area that radiates into the buttock, thigh, and legs, causing or mimicking sciatica symptoms. It was quantified as presence or absence. Gastroesophageal reflux is heartburn and was quantified as presence or absence. Both detail were included (lines 123-125).

Line 127-vaginal tear - to be more specific grade 1, grade 2 etc specifically grad 3 and 4?

Response: vaginal tear refers to grades 3 or 4, which is considered severe vaginal tears. They are more frequent in adolescents due to tightness of the vagina. This was included in line 142.

Line 137- 138 and 210 -it must be point out that EPDS it is not an diagnostic tool but a screening tool so you must introduce the results in that context.

Response: We agree with this comment and this has now been clarified in the main text.

Line 131- loss of fetal well-being -must be more precisely point out -variable decelerations, late decelerations, bradycardia, etc.?  It is detected by the non-stressed test, including alterations in heart rate, variable decelerations, late decelerations, bradycardia.

Response: The fetal well-being was categorized according the umbilical artery Doppler velocimetry screening test for high-risk population. This detail was included in line 146.

Finally, something must be point out in table 2 and table 3. There are some confusing data. line 178 and line 190 point out something and in In table 2 the lower back pain is increased >17 years old and not decreased, anemia is also increased in > 17 y old not decreased, and in table 3 the FGR is increased  in > 17 y old patients, SGA is also increased and not decreased in > 17 y old patient and overall malformations the same. So it must be detailed why are these results presented because also the p results at table 3 are confusing.

Response: We believe that the reviewer's confusion is produced by the fact that we have used two analysis. On the one hand, categorizing women according to their maternal age and establishing associations by contingency tables. This is interesting considering the high risk of obstetric complications in women with pregnancies in the first 5 years after menarche. The prevalence is not significant in any case.

In the second approach, we used regression models to test the prevalence of risk (increased/decreased) for each year increase of maternal age (ORs). Using this approach we found several significant differences. The variation in obstetric complications was established by OR>1 or OR<1.

Reviewer 2 Report

Thank you for the opportunity to review this manuscript. It brings an important topic for public health, but I believe it needs several edits to be suitable for publication. Below, I provide suggestions and questions for each of the paper’s sections.

Abstract:

The abstract should start with the information that justifies the study. I mean, yes, adolescent pregnancy is a health concern worldwide, but why studying specifically the impact of maternal age among adolescents on maternal and child complications is important?

Introduction:

Please, include references for the sentences in lines 51-52 and 53-54.

The introduction needs more information about the justification of the study. The authors explain the impact of adolescent fertility on maternal and child’s health, but there is insufficient information about the importance of investigating the importance of the adolescents’ age on that complication, what is already know about it and why studying this is still necessary.

Methods:

Please provide more information about the HULP. I am more interested in knowing if this hospital assists patients with health insurance or population in general. I am not familiar with the health system in Spain and more information about this may help the reader to understand the participant’s profile.

Regarding the eligibility criteria, if 13 to 19 years was considered as inclusion criteria, those with age over 20 were not included, so this is not an exclusion criterion.

How many participants were excluded because of incomplete medical data and because of sexually transmitted infection or toxic habits? It is important to know how many adolescents gave birth at this hospital during the study data collection and how many of them participated of the study.

Please, provide more information about the data collected. Did all participants have a standard medical record with the same questions to be answered by the health professional? How was this information collected? Was it collected when the participant was checked in the hospital for delivery or during prenatal care? Explain more about the origin of this medical record in the text, please. I am more concerned in understanding how reliable these data are. If the medical form is not standard, it is possible that some information was simply not recorded, although the participant may have had it.

Please, explain in the statistics that the regression models were built to test the contribution of increasing one year in maternal age on each complication.

Results:

The authors say that Rumanian was the largest group, but there is no information about this classification in the methods. Also, Rumanian seems to be the largest group among the Caucasian, and Latin-American origin is a completely different group. The way is written, is seems the authors are saying that Latin-American is the second among the Caucasian. Just rephrase it to let it clearer.

Table 1 would benefit of more information about the sample characteristics that the authors cite when discussing the results. For instance, the mean age at menarche in each group, or classification of socioeconomic status. If the authors did not collect this information, it should be commented and considered as a limitation of the study.

The authors say that there were no miscarriages in the cohort. If giving birth at the study hospital was an inclusion criterion, I would not expect any miscarriages in the sample, although stillbirths could be possible. Please, explain if this is correct and clarify in the text.

The authors report they divided the participants in two groups of age, below 17 and over 17. And 17 years old participants were included in which group?

Line 175: Please, explain that 0.9 refers to the OR.

Please, include in the footnote of the tables that the analyses of the regression model were made considering age as a continuous variable.

Line 187: Table 3 “shows”.

Please explain how the variables Maternal-obstetrical complications, fetal complications, etc., were operationalized. If a participant had more than one complication, it was considered only once for that group? Or the number of complications reported by participants of that group was considered to determine those variables?

Discussion:

Please check the spelling in the sentence around lines 238-239.

Lines 250-241: please correct Spanish in women by Spanish women.

Regarding the discussion about the FGR and SGA, the authors explain these rates may be higher because 11% of the pregnancies recorded were not under obstetrical control. Does this mean that you do not have information about FGR and SGA of those participants? If yes, do not you think those participants should be excluded from the analysis in which the information was not recorded? Moreover, this falls in what I asked before about how standardized the medical records are. If there is no information about one complication, it does not necessarily mean that it did not occur, since it may not have been recorded. This is an important source of bias and must be discussed. I suggest the authors comment about this in the limitations section.

About the route of delivery, the authors say that they detected more vaginal deliveries than C-sections and contrasted it with another study that said that adolescents are at higher risk of C-section and forceps. It is not clear how both studies are different because it seems they presented different perspectives. I mean, it seems that the study you cited found that adolescent pregnancies present higher risk of C-section than adults, which is different of saying that most adolescents have C-sections. Please, explain better what you mean.

The authors say “Surprisingly and overall, complications during labor increased with maternal age. This could be explained by the higher rate of prematurity in our sample (11.1%) compared to the general population [26-29]”. I am not sure what the authors mean. How can higher rates of prematurity in the entire sample explain the increase of complications during labor with increased maternal age?

Line 330-332 – The authors say “Our data demonstrate the risk of maternal-fetal complications in adolescents, higher in those of younger age. We also showed that they are higher in non-Spanish origin, likely with lower socioeconomic and educational level”. Do you mean that maternal-fetal complications are higher in non-Spanish adolescents? I did not find these results in our study. Please clarify.

Lines 333-339 – The authors comment the need to explore the influence of socioeconomic level, stress, and conflicts to reduce “this medical issue”. Do the authors refer to the complications or to the adolescent pregnancy itself? It seems these sentences are more related to how preventing adolescent pregnancies, but I think that the authors should mention how to reduce the complication among those who are already pregnant, given this paper objectives.

Conclusion:

I believe the conclusion should also include more information about the implications of these results for those adolescents who are already pregnant. From my perspective, I believe the great message from this study is that younger adolescent mothers are at even higher risk of complications. Thus, health strategies to reduce the risks for them must be implemented, such as improving their prenatal assistance or other specific policies recognizing their higher risk.

Author Response

Thank you for the opportunity to review this manuscript. It brings an important topic for public health, but I believe it needs several edits to be suitable for publication. Below, I provide suggestions and questions for each of the paper’s sections.

Response: We would like to thank the reviewer for the time spent on our manuscript as well as for the comments and suggestions made. We hope that we have kindly answered all the queries.

Abstract: The abstract should start with the information that justifies the study. I mean, yes, adolescent pregnancy is a health concern worldwide, but why studying specifically the impact of maternal age among adolescents on maternal and child complications is important?

Response: We focused on maternal age, since there are some, but scarce data suggesting that women within the first 5 years of menarche have higher risk of complications. We have clarified the main aim and justified it in the abstract.

Introduction:

Please, include references for the sentences in lines 51-52 and 53-54.

Response: The reference [8] was included at the end of the paragraph.

The introduction needs more information about the justification of the study. The authors explain the impact of adolescent fertility on maternal and child’s health, but there is insufficient information about the importance of investigating the importance of the adolescents’ age on that complication, what is already know about it and why studying this is still necessary.

Response: Data from developing countries indicate that fetal and obstetric complications are higher in adolescents with ages near menarche. Poor socioeconomic conditions or health systems may exert an influence. Therefore, we aimed to evaluate this aspect in Spain, a country with a higher socioeconomic situation and with a well-established Public Health System. We have clarified this aspect in the introduction, including reference 16 and 17, and we have added some information regarding the risks that adolescent pregnancies have for maternal and fetal health.

Methods:

Please provide more information about the HULP. I am more interested in knowing if this hospital assists patients with health insurance or population in general. I am not familiar with the health system in Spain and more information about this may help the reader to understand the participant’s profile.

Response: In Spain there is a well-established public health system, accessible to everyone. The majority of people are attended by this system, although in recent years there has been an expansion of privatization of health. Despite this fact, the public hospitals as HULP, are a reference high-quality hospitals. It is a referral hospital in Madrid (Spain), which assists general population. It attends population mostly from Madrid, but patients also derived from other cities for specific services. The Obstetric Service at HULP attends an average of 5,500 deliveries per year.

Regarding the eligibility criteria, if 13 to 19 years was considered as inclusion criteria, those with age over 20 were not included, so this is not an exclusion criterion.

Response: To avoid possible confusion, "maternal age over 20" was removed from the main text.

How many participants were excluded because of incomplete medical data and because of sexually transmitted infection or toxic habits? It is important to know how many adolescents gave birth at this hospital during the study data collection and how many of them participated of the study.

Response: Considering the reviewer's comment, we have included the number of pregnancies excluded due to concomitant sexually transmitted disease, pregnancies with confirmed toxic habits and the ratio of annual adolescent births attended at the time of data collection.

Please, provide more information about the data collected. Did all participants have a standard medical record with the same questions to be answered by the health professional? How was this information collected? Was it collected when the participant was checked in the hospital for delivery or during prenatal care? Explain more about the origin of this medical record in the text, please. I am more concerned in understanding how reliable these data are. If the medical form is not standard, it is possible that some information was simply not recorded, although the participant may have had it.

Response:  In Spain, prenatal care starts in Primary Health Centers and, in the case of high-risk pregnancies, they are appointment to their zone hospital (i.e. HULP) for follow-up. At the hospital, women are assigned an obstetrician who decides the number of visits and registers the data in her medical record, except the data obtained during labor. In the case of adolescent pregnancies, since they are high risk pregnancies, women are directly derived to the hospital for follow-up. However, some of women do not even go to their primary health centers, due to fear or ignorance and they go to the hospital to delivery. In this study we had some of these women, and we could only collect data at labor. We have included some more information about this aspect (lines 107-111).

Please, explain in the statistics that the regression models were built to test the contribution of increasing one year in maternal age on each complication.

Response: This information was included in the text.

Results:

The authors say that Rumanian was the largest group, but there is no information about this classification in the methods. Also, Rumanian seems to be the largest group among the Caucasian, and Latin-American origin is a completely different group. The way is written, is seems the authors are saying that Latin-American is the second among the Caucasian. Just rephrase it to let it clearer.

Response: The most representative ethnic group was Caucasian, being Rumanian the largest cluster. The second ethnic group was formed by women of Latin American origin. We have rephrased the sentence to make it clear.

Table 1 would benefit of more information about the sample characteristics that the authors cite when discussing the results. For instance, the mean age at menarche in each group, or classification of socioeconomic status. If the authors did not collect this information, it should be commented and considered as a limitation of the study.

Response: Unfortunately, these characteristics were not collected. This was included as a limitation of the study in the discussion.

The authors say that there were no miscarriages in the cohort. If giving birth at the study hospital was an inclusion criterion, I would not expect any miscarriages in the sample, although stillbirths could be possible. Please, explain if this is correct and clarify in the text.

Response: It is possible that during the follow-up of the pregnancy, stillbirths could be detected, which would be included in the medical record. However, we did not have any cases in the women included in this study. We have changed the terminology to avoid confusions.

The authors report they divided the participants in two groups of age, below 17 and over 17. And 17 years old participants were included in which group?

Response: We have clarified this aspect in the material and methods. The categories were less than 17 years (<17) and equal to 17 years or older (≥17).

Line 175: Please, explain that 0.9 refers to the OR.

Response: this explanation was added.

Please, include in the footnote of the tables that the analyses of the regression model were made considering age as a continuous variable.

Response: this explanation was added.

Line 187: Table 3 “shows”.

Response: Corrected. Thank you

Please explain how the variables Maternal-obstetrical complications, fetal complications, etc., were operationalized. If a participant had more than one complication, it was considered only once for that group? Or the number of complications reported by participants of that group was considered to determine those variables?

Response: the calculation was made considering the diagnosis of some of them. This methodology was previously published (PMID: 27780538) to determine the cumulative effect. This was explained in the material and methods.

Discussion:

Please check the spelling in the sentence around lines 238-239.

Response: The sentence was corrected.

Lines 250-241: please correct Spanish in women by Spanish women.

Response: Corrected.

Regarding the discussion about the FGR and SGA, the authors explain these rates may be higher because 11% of the pregnancies recorded were not under obstetrical control. Does this mean that you do not have information about FGR and SGA of those participants? If yes, do not you think those participants should be excluded from the analysis in which the information was not recorded? Moreover, this falls in what I asked before about how standardized the medical records are. If there is no information about one complication, it does not necessarily mean that it did not occur, since it may not have been recorded. This is an important source of bias and must be discussed. I suggest the authors comment about this in the limitations section.

Response: we agree with this comment. For this reason, we have included this limitation in the main text in the discussion section.

About the route of delivery, the authors say that they detected more vaginal deliveries than C-sections and contrasted it with another study that said that adolescents are at higher risk of C-section and forceps. It is not clear how both studies are different because it seems they presented different perspectives. I mean, it seems that the study you cited found that adolescent pregnancies present higher risk of C-section than adults, which is different of saying that most adolescents have C-sections. Please, explain better what you mean.

Response:  You are right, the reference refers to a comparison between adolescents and non-adolescent pregnancies and we are not comparing this. The sentence was confusing, and we have rephrased it.

The authors say “Surprisingly and overall, complications during labor increased with maternal age. This could be explained by the higher rate of prematurity in our sample (11.1%) compared to the general population [26-29]”. I am not sure what the authors mean. How can higher rates of prematurity in the entire sample explain the increase of complications during labor with increased maternal age?

Response: Both prematurity (which is associated with C-section) and vaginal tear (added) could increase the sum of complications during labor. These three complications are very prevalent in adolescent pregnancies. We detected a trend association between prematurity and maternal age. Although it was not significant. We have reformulate the text to clarify it (Lines 331-335).

Line 330-332 – The authors say “Our data demonstrate the risk of maternal-fetal complications in adolescents, higher in those of younger age. We also showed that they are higher in non-Spanish origin, likely with lower socioeconomic and educational level”. Do you mean that maternal-fetal complications are higher in non-Spanish adolescents? I did not find these results in our study. Please clarify.

Response: Apologies for the mistake. This is due to the fact that the majority of our cohort were women was of non-Spanish origin (mostly from Romania and Latin American countries). We have clarified this sentence in the text.

Lines 333-339 – The authors comment the need to explore the influence of socioeconomic level, stress, and conflicts to reduce “this medical issue”. Do the authors refer to the complications or to the adolescent pregnancy itself? It seems these sentences are more related to how preventing adolescent pregnancies, but I think that the authors should mention how to reduce the complication among those who are already pregnant, given this paper objectives.

Response: This reviewer's appreciation is very interesting. We wanted to keep both points of view in the discussion, 1) reinforcing health policies to prevent adolescent pregnancies and 2) intervening when they have already occurred.

Conclusion:

I believe the conclusion should also include more information about the implications of these results for those adolescents who are already pregnant. From my perspective, I believe the great message from this study is that younger adolescent mothers are at even higher risk of complications. Thus, health strategies to reduce the risks for them must be implemented, such as improving their prenatal assistance or other specific policies recognizing their higher risk.

Response: We agree on the comment and we have included this aspect in general conclusion.

Reviewer 3 Report

I recommend the following suggestion for further improving the manuscript. Methods: Please mention about how you excluded stillbirths or termination pregnancies. If these were part of exclusion criteria, this needs to be mentioned. The rates of adolescent pregnancies are reported incorrectly. Please correct the order of numerator and denominator in the parentheses. What is the rationale for separating the age groups

Author Response

Response: We would like to thank the reviewer for the time spent on our manuscript as well as for the comments and suggestions made. We hope that we have kindly answered all the queries.

I recommend the following suggestion for further improving the manuscript.

Methods: Please mention about how you excluded stillbirths or termination pregnancies. If these were part of exclusion criteria, this needs to be mentioned.

Response: There were not miscarriages in this study. We have changed miscarriage for stillbirths in the result section to avoid confusions.

The rates of adolescent pregnancies are reported incorrectly. Please correct the order of numerator and denominator in the parentheses.

Response: Corrected. Thank you.

What is the rationale for separating the age groups?

Response: We established these two groups considering high risk of obstetric complications in women with pregnancies in the first 5 years after menarche, assuming an average menarche age of 11 years (PMID: 18974233).

Reviewer 4 Report

Overall a nicely written paper. Primary comments/suggestions are to clarify the use of rates versus proportions in statistical analysis, and better align discussion and conclusions based on your primary analysis. See details below.

1) Overall comment on data presentation: Improve description of rates versus proportions. Reference fertility rates (per 1,000 adolescent women) but most "rates" presented are proportions (%). Therefore it is not evident what the denominator is for calculating proportions referenced in the text and Tables. For example, line 68, lines 90-92, proportions in Tables 1-4. Also don't understand when the Tables reflect proportion of each adolescent birth versus proportion of all complications (?). 

2) Under Data Collection: a) define obstetric control. How many prenatal visits, were all deliveries attended, duration of prenatal or postnatal care - what is the criteria for obstetric control? Other authors have stratified # prenatal visits to reflect higher or lower level of prenatal care; b) what was the minimum criteria for available medical records to be included in the study sample?

3) Under Results/statistical analysis: a) overall, a high % of pregnancies were to healthy adolescents with no comorbid conditions; b) most pregnancy and delivery outcomes were uncomplicated regardless of age (except for labor complications); c) In Table 2, except for hyperemesis, the statistically significant differences between groups were minimal in that both age groups experienced low 'rates' of complications - this challenges how to interpret statistical significance, which does not necessarily reflect clinically relevant significance between age groups; d) Table 3 - the 'rates' of malformations were higher among older adolescents, yet the text (lines 187-192) states that rates of malformations decreased with age. e) Table 4 - labor complications seemed high, although it was not clear what the total percentage by age group represented (i.e., number of unique adolescents with any of the listed complications? how many adolescents experienced >1 complication?); f) breastfeeding intention data adds little to the paper, especially with no explanation of how that data were collected.

4) Discussion/study limitations/conclusions: a) the study sample might not be representative of all adolescent pregnancies in Spain, given the high % of non-Latin ethnicity among study group participants; pregnancy care in HULP might also not be reflective of pregnancy care outside of large medical center in Madrid; b) good that authors indicated the potential biases related to ethnicity; c) there was a lot of discussion referencing the impact of low socioeconomic factors yet that indicator was not included in the analysis of your study population; d) what is the definition of a 'high income country'? e) several paragraphs describe the important of access to sexual health education, contraception, socioeconomic or sociocultural factors. However, the discussion seems to overinterpret the study findings as the focus was on age as the primary risk factor for poorer pregnancy outcomes. The authors did not show crude data stratified by ethnicity nor present data collected directly from the study population about these other factors; f) be careful about conclusions when referencing rates of rare pregnancy outcomes; g) a lot of discussion about postpartum depression even though no study participant received that diagnosis; h) good for authors to acknowledge the limitation of excluding adolescents with a history of smoking, alcohol or illegal drug use.

Author Response

Overall a nicely written paper. Primary comments/suggestions are to clarify the use of rates versus proportions in statistical analysis, and better align discussion and conclusions based on your primary analysis. See details below.

Response: We would like to thank the reviewer for the time spent on our manuscript as well as for the comments and suggestions made. We hope that we have kindly answered all the queries.

Overall comment on data presentation: Improve description of rates versus proportions. Reference fertility rates (per 1,000 adolescent women) but most "rates" presented are proportions (%). Therefore it is not evident what the denominator is for calculating proportions referenced in the text and Tables. For example, line 68, lines 90-92, proportions in Tables 1-4. Also don't understand when the Tables reflect proportion of each adolescent birth versus proportion of all complications (?).

Response: We agree with the reviewer. To avoid possible confusion, all ratios have been converted to %. In the case of the tables, the % were calculated with the sample size of each complication (represented between parentheses) out total of the category (shown in the column heading with "n").

Under Data Collection:

  1. Define obstetric control. How many prenatal visits, were all deliveries attended, duration of prenatal or postnatal care - what is the criteria for obstetric control? Other authors have stratified # prenatal visits to reflect higher or lower level of prenatal care;

Response: Regular obstetric visits or obstetrical control means one visit a month, with the obstetrician or the midwife. Out of obstetric control refers to those pregnant women who did not have prenatal care and only went to the hospital for the labor. Labor assistance means that labor was assisted at the hospital. All the deliveries were attended at the hospital. These details were incorporated in the main text.

  1. What was the minimum criteria for available medical records to be included in the study sample?

Response: The minimum clinical history data to be included in the study were maternal age, origin and clinical data of delivery.

Under Results/statistical analysis: a) overall, a high % of pregnancies were to healthy adolescents with no comorbid conditions; b) most pregnancy and delivery outcomes were uncomplicated regardless of age (except for labor complications); c) In Table 2, except for hyperemesis, the statistically significant differences between groups were minimal in that both age groups experienced low 'rates' of complications - this challenges how to interpret statistical significance, which does not necessarily reflect clinically relevant significance between age groups;

Response: The reviewer's assessments are correct. Fortunately, the prevalence of these complications was low. Nevertheless, although not clinically relevant, it is important to identify which complications may occur more frequently in adolescent pregnancies.

  1. Table 3 - the 'rates' of malformations were higher among older adolescents, yet the text (lines 187-192) states that rates of malformations decreased with age. e) Table 4 - labor complications seemed high, although it was not clear what the total percentage by age group represented (i.e., number of unique adolescents with any of the listed complications? how many adolescents experienced >1 complication?);

Response: We believe that the reviewer's confusion is produced by have used two analysis approach. On the one hand, categorizing women according to their maternal age and establishing associations by contingency tables. This is interesting considering the high risk of obstetric complications in women with pregnancies in the first 5 years after menarche. The prevalence is not significant in any case.

In the second approach, we established regression models to test how the prevalence of risk increased/decreased for each year increase of maternal age (ORs) and evidenced some statistical differences. This variation in obstetric complications is established by OR>1 or OR<1.

  1. Breastfeeding intention data adds little to the paper, especially with no explanation of how that data were collected.

Response.  It is true that the evaluation of breastfeeding seems to be away from the general objective of the manuscript, focused on the risk of complications. However, we considered important to evaluate it since, exclusive breastfeeding for at least the first 6 months of postnatal age is a WHO gold standard. We aimed to assess the influence of maternal age since other studies have shown that adolescent pregnancies have low breastfeeding adherence. This may be due to the fact that they experience more pain in the process, as well as low psychological capital to handle breastfeeding, which may affect the deterioration of the maternal-filial bond. Breastfeeding is a health indicator for both mother and newborn, and prevention and intervention policies should detect women potentially at risk of breastfeeding cessation. This information was collected by asking directly to the mother.

Discussion/study limitations/conclusions: a) the study sample might not be representative of all adolescent pregnancies in Spain, given the high % of non-Latin ethnicity among study group participants; pregnancy care in HULP might also not be reflective of pregnancy care outside of large medical center in Madrid;

Response: HULP is a national reference hospital in obstetrics, although other geographical areas may have other factors, this could represent what happens in a multicultural city like Madrid in a developed country. The data show a social fact, that adolescent pregnancies are higher in social groups of non-Spanish origins. This is a reality in our society and an important aspect to take into account to implement policies to reduce adolescent pregnancies. We have discussed these aspects further.

  1. good that authors indicated the potential biases related to ethnicity; c) there was a lot of discussion referencing the impact of low socioeconomic factors yet that indicator was not included in the analysis of your study population;

Response: We agree with this comment, which is why socioeconomic factors were included as limitations of the study. However, it would be impossible to discuss these data without mentioning the social context in which they are produced, which is influenced by economic and educational factors.

  1. What is the definition of a 'high income country'?

Response: The World Bank classifies as high-income those countries with Gross National Product (GNP or Income, GNI) per capita income of $9,266 or more in 2000 (https://stats.oecd.org/glossary/detail.asp?ID=1231).

  1. Several paragraphs describe the important of access to sexual health education, contraception, socioeconomic or sociocultural factors. However, the discussion seems to over interpret the study findings as the focus was on age as the primary risk factor for poorer pregnancy outcomes. The authors did not show crude data stratified by ethnicity nor present data collected directly from the study population about these other factors;

Response: It is true that socioeconomic factors related to poorer ethnic groups may have influenced the results and need to be taken into consideration, as included in limitations. However, the fact is that we found a significant impact of maternal age on the complications related to adolescent pregnancies. We have now discussed separately the medical and social aspects, which before were discussed together, possible causing confusion.

  1. be careful about conclusions when referencing rates of rare pregnancy outcomes; g) a lot of discussion about postpartum depression even though no study participant received that diagnosis;

Response: We wanted to clarify some points of the discussion, being more cautious with conclusions. On the other hand, we thought it necessary to include postpartum depression since adolescent pregnancies may have an increased risk. We have used EPDS as a screening tool and not as diagnosis and we have mentioned it in text.

  1. Good for authors to acknowledge the limitation of excluding adolescents with a history of smoking, alcohol or illegal drug use.

Response: Thanks for the observation. We are aware of this limitation. The data of this article may serve as a starting point and it would be interesting to continue further studies, taking into account this important factors, which are likely influencing the medical outcome.

Round 2

Reviewer 2 Report

Thank you for providing a review of the paper. It improved since the first version, but I still think it needs minor changes.

Introduction:

“Regarding maternal complications, higher rates of anemia, pregnancy-induced hypertension and preeclampsia have been reported. There is also evidence of higher rates of fetal complications, such as fetal growth restriction (FGR), prematurity, miscarriage, and fetal death.” – these two sentences need the proper citation. In that case, you must cite the papers that found those complications. Include a citation at the end of the paragraph is not enough, in my opinion.

Methods:

About the participants included, if I understood correctly, only 297 adolescents gave birth at this hospital during the study period (279 included and 18 excluded). Is that correct? If yes, just add a simple sentence saying the population (N=297), so the reader can understand better how representative this sample is.

Regarding the complications, I still do not understand how they were categorized. Reading the table, I understood that this refers to a categorical variable, with some participants being categorized as having those complications and other do not. However, the authors just said, “Cumulative maternal-obstetrical complications was calculated as a sum of all above diagnoses”. But this did not explain how variable was categorized as “yes” or “no”. Please, clarify in the text.  Also, check the grammar here and in the other sentences – cumulative complications “were”.

Line 368 – exclude the comma after “implement”.

I had suggested to include the information about age at menarche in the study, but the authors answered they had not collected it. In that case, don’t you think you should add at least a comment about this? In the introduction you suggested that being pregnant near the menarche is a possible explanation for the worst results among the younger adolescents, right? So, although is more likely that the older group have had the menarche more years before the younger group, this is just a supposition. I mean, it is possible that in both groups the periods between age at menarche and the pregnancy are similar if women from the older group had an older age at menarche than the younger group. Without the information about the age at menarche, you cannot test whether the results are related to the participants’ age at pregnancy or the period between pregnancy and menarche. If you agree with that, you should add a comment about this, perhaps in the limitations.

Author Response

Thank you for providing a review of the paper. It improved since the first version, but I still think it needs minor changes.

Response: Thank you for your time to review carefully our article. Below, we have provided a responses point-by-point to your comments.

Introduccion: “Regarding maternal complications, higher rates of anemia, pregnancy-induced hypertension and preeclampsia have been reported. There is also evidence of higher rates of fetal complications, such as fetal growth restriction (FGR), prematurity, miscarriage, and fetal death.” – these two sentences need the proper citation. In that case, you must cite the papers that found those complications. Include a citation at the end of the paragraph is not enough, in my opinion.

Response: We have included appropriate cites [8, 9, 10] in each sentence.

Methods: About the participants included, if I understood correctly, only 297 adolescents gave birth at this hospital during the study period (279 included and 18 excluded). Is that correct? If yes, just add a simple sentence saying the population (N=297), so the reader can understand better how representative this sample is.

Response: Yes, this is correct. We have clarified the final sample size (lines 95-96).

Regarding the complications, I still do not understand how they were categorized. Reading the table, I understood that this refers to a categorical variable, with some participants being categorized as having those complications and other do not. However, the authors just said, “Cumulative maternal-obstetrical complications was calculated as a sum of all above diagnoses”. But this did not explain how variable was categorized as “yes” or “no”. Please, clarify in the text.  Also, check the grammar here and in the other sentences – cumulative complications “were”.

Response: We understand the confusion. Women with maternal, fetal and labor complications were categorized as yes/no. We did not perform any statistical analysis regarding cumulative sum of complications. Therefore, we have eliminated of the text this aspect and modified it to clarify the categorization.  

Line 368 – exclude the comma after “implement”.

Response: corrected.

I had suggested to include the information about age at menarche in the study, but the authors answered they had not collected it. In that case, don’t you think you should add at least a comment about this? In the introduction you suggested that being pregnant near the menarche is a possible explanation for the worst results among the younger adolescents, right? So, although is more likely that the older group have had the menarche more years before the younger group, this is just a supposition. I mean, it is possible that in both groups the periods between age at menarche and the pregnancy are similar if women from the older group had an older age at menarche than the younger group. Without the information about the age at menarche, you cannot test whether the results are related to the participants’ age at pregnancy or the period between pregnancy and menarche. If you agree with that, you should add a comment about this, perhaps in the limitations.

Response: Thank you so much for this interesting comment. We agree with that and we have included this point of view in the limitation of the study (lines 401-405).